# Molecular detection and antibiotic resistance of diarrheagenic *Escherichia coli* from street food and water in mukuru slums, Nairobi County

Sheillah Mundalo[1,2*], Regina Ntabo[2], Kelvin Kering[1], Rael Too[1], Kevin Kariuki[1], Diana Imoli[1], Brian Silantoi[1], Evans Kiptanui[1], Susan Kavai[1], Samuel Kariuki[1] Cecilia Mbae[1]

1 Centre for Microbiology Research, Kenya Medical Research Institute, Nairobi, Kenya, 2 Department of Biotechnology, Biochemistry and Microbiology, Kenyatta University, Nairobi, Kenya

* smundalo@kemri.go.ke

## Abstract

Globally, diarrheal diseases account for 550 million cases of foodborne illnesses annually. In Kenya, *Escherichia coli* (*E. coli*) infections from contaminated food and water pose a serious health concern, especially in settings with poor sanitation and hygiene practices This study examined the genetic characteristics and antimicrobial resistance profiles of diarrheagenic *E. coli* (DEC) recovered from street foods and water from Mukuru informal settlements, Nairobi. Between September and December 2023, 384 (each 77) samples of street foods (*Mandazi, githeri, French fries*), wastewater, and drinking water were collected and *E. coli* isolation performed through microbiological culture and antibiotic susceptibility testing done using Kirby-Bauer disc diffusion method. Conventional Polymerase chain reaction (PCR) was used to screen for six DEC and extended-spectrum beta-lactamase (ESBL) resistance genes. Descriptive and inferential statistics (Pearson's chi-square test) were used to assess associations between sample types, *E. coli* positivity, pathotypes, and antibiotic resistance. *E. coli* was isolated in 16% (62/384) of the samples, with 77.4% (48/62) of the isolated *E. coli* being DEC. Majority (64.6%, 31/48) of DEC isolates were recovered from wastewater followed by drinking water (22.9%, 11/48), *githeri* (8.3%, 4/48), *mandazi* (4.2, 2/48), and none from *french fries*. The most common pathotypes were; ETEC 69% (33/48), STEC 52.1% (25/48), EIEC 50% (24/48)), EPEC 10% (5/48), and EAEC 6% (3/48) Of the 48 DEC isolates, 30 were hybrid isolates Overall, the *E. coli* isolates were highly resistant to tetracycline (77.4%), trimethoprim-sulfamethoxazole (71.0%), ampicillin (59.7%) and least resistant to kanamycin (16.1%); chloramphenicol (8.1%) and amoxicillin + clavulanic acid (4.8%). A huge proportion (72.6%, 45/62) of the isolated *E. coli* were multidrug-resistant (MDR). Of the 45 MDR isolates, 60% (27) were from wastewater, 28.9% (13) from drinking water, 4.4% (2) from githeri, 4.4% (2) from mandazi and 2.2% (1) french fries 2.2%. ESBL genes *bla*-$_{TEM}$ and *bla*$_{CTX-M}$

**Data availability statement:** All relevant data neede to replicate the analysis are within the manuscript and its Supporting Information files.

**Funding:** The author(s) received no specific funding for this work.

**Competing interests:** The authors have declared that no competing interests exist.

were detected in 12.9% and 3.2% of the *E. coli* isolates. isolates. The high prevalence of MDR in the *E.coli* isolates recovered from environmental compartments and food is a huge public health risk to the population in these settings. The detection of *E. coli* indicates fecal contamination underscoring the need to improve water, and sanitation infrastructure in urban slums.

## Introduction

Food and water safety is an important component of public health and economic prosperity. A large proportion of contaminated food results in diarrheal diseases, with diarrheal disease agents accounting for 550 million of the 600 million cases of foodborne illness [1]. Each year, an estimated 600 million people fall ill and 420,000 people die from unsafe food [2]. Global Burden of Foodborne Diseases report estimated that 111 million illnesses and 63,000 deaths are caused by diarrhoeagenic *E. coli* globally each year [3]. Low- and middle-income countries (LMICs) bear the highest burden of diarrheal disease, predominantly caused by *E. coli* [4]. The sale of ready-to-eat street food represents an important source of income in many developing countries, however, studies have highlighted street foods as a major public health risk [5].

In Kenya, diarrheal diseases rank as the third most significant cause of pediatric morbidity and mortality, after malaria and tuberculosis [6]. *E. coli* is a major contributor to ill health, including neonatal meningitis, urinary tract infections (UTIs), pyelonephritis, cystitis, septicemia, and traveler's diarrhea [7]. Exposure to *E. coli* most commonly occurs through ingesting contaminated food and water. Previous studies have indicated that pathogenic strains of *E. coli* develop from commensal/naïve bacteria by acquiring specific virulence genes and operons either on the chromosome or through extra-chromosomal elements [8]. Diarrheagenic *E. coli* is categorized into five distinct pathotypes based on their clinical manifestation' phenotypic traits, epidemiological evidence, and virulence factors specific to each pathotype [9]. They include; Enteroaggregative (EAEC), Shiga-toxigenic (STEC), Enteroinvasive (EIEC), Enteropathogenic (EPEC) and Enterotoxigenic (ETEC). Antibiotic treatment is the primary method for preventing and managing bacterial infections. However, the growing antibiotic resistance among bacterial pathogens transmitted through food poses a major challenge, leading to increased mortality and morbidity from foodborne illnesses resulting in high socioeconomic costs [10]. Different studies conducted on different items of ready-to-eat foods vended on the street in low-income countries, revealed different levels of contamination and the majority of foodborne bacterial pathogens were resistant to commonly prescribed antibiotics [11]. The emergence of antimicrobial resistance has led to challenges in the management of infectious diseases that affect both animals and humans [12]. The World Health Organization has ranked *E. coli* as the second most important pathogen globally that is resistant to antimicrobials [13]. The spread of antimicrobial resistance in pathogenic strains of *E. coli* is critical because infections caused by these strains are often harder to treat,

resulting in increased severity and duration of infection [14]. One important group of antimicrobial-resistant *E. coli* is the ESBL-producing *E. coli* [15]. Since bacteria are highly prone to exchanging genetic material, these resistance genes are easily disseminated between bacteria [16].

The rise of multidrug-resistant *E. coli* strains poses a significant public health challenge and complicates the management of serious infections [17]. In Kenya, there is evidence of multidrug-resistant *E. coli* isolated from clinical and environmental samples [18]. The prevalence of diarrheagenic *E. coli* (DEC) and its resistance profiles from street food and water is not well known especially in low-resource settings such as slums. The present study sought to determine the occurrence, distribution and AMR profiles of DEC pathotypes in street foods and water samples from Mukuru informal settlement. This study investigated multidrug-resistant *E. coli* and the prevalence of pathogenic *E. coli* isolated from street food and water samples in the Mukuru informal settlement in Nairobi.

## Materials and methods

### Study site and design

This was a cross-sectional study where samples were randomly collected from food vendors and water samples from boreholes in Mukuru Informal settlement. Mukuru is located 20 km East of Nairobi city center. With a population of approximately 150,000, it covers an area of approximately 2 km² (population density, 75,000/km²) [19]. Only 1% of residents in Mukuru have access to a private or individual water source. There is an average of 234 households per public water tap in Mukuru and the households are water insecure, since they have access to 1.8 cubic meters of water per month, while the international recommendation for health and hygiene is 4 cubic meters per month [20].

### Sample collection

We standardized the SaniPath protocol [21] that has been applied in exposure assessment to obtain our samples (S1 Fig) [21]. After obtaining verbal consent, we collected the street food vendors prepared as a single serving and aseptically poured/placed the food into a 500 mL Whirl-Pak bag. Drinking water was collected from communal municipal water points into a 500 mL Whirl-Pak bag while wastewater sampling points were randomly selected based on areas where human interaction was most frequent. The study employed the Fischer formula $n = Z^2PQ/d^2$ to determine a sample size of 384 where n is the estimated sample size, Z is the Z-score corresponding to the desired confidence level (1.96 for a 95% confidence level), P is the estimated proportion (0.50 or 50% in this case since no previous estimate is available), Q is the complementary probability to P, which is equal to 1 -P and d is degree of accuracy (0.05 at 95% confidence level)

Samples were then aseptically transported to the KEMRI microbiology laboratory in cooler boxes and processed within six hours of sample collection.

**Laboratory analysis.** Samples from street foods were rinsed in distilled water and then a loopful of the 10 mL of rinse solution was cultured in Chromocult® Coliform Agar (Millipore at # ISO 9308−1). Water samples were cultured directly in Chromocult® Coliform Agar. After 18 hours of incubation, *E. coli* suspects (blue colonies) were sub-cultured on Muller Hilton agar and incubated for 18hrs. *E. coli* identification was done using analytical profile index (API) 20E strips (bioMerieux, Inc., Durham, North Carolina, United States). DNA was extracted from pure *E. coli* colonies using the boiling method [22]. Briefly, a loop full of the colony was suspended in 200 µl of PCR water in an Eppendorf tube and heated at 95°C or 8 minutes on a heating block. Centrifugation was then done at 15,000g for 5 minutes at 25°C. The DNA-containing supernatant was aliquoted to a sterile Eppendorf tube.

**Detection of *E. coli* pathotypes.** Multiplex polymerase chain reaction assays were used for the detection of five types of DEC as previously described [23]. Multiplex was done in EPEC, ETEC, and EIEC while singleplex was done on EAEC and STEC. The targeted genes for the pathogenic *E. coli* strains (**Table 1**) included ETEC (*lt* encoding heat-labile enterotoxins), STEC (*stx₁* encoding Shiga-toxin), EPEC (*bfp* encoding bundle forming pili), EAEC (*aggR* encoding transcription regulator

**Table 1.** *Escherichia coli* pathotypes genes analyzed and the primers.

| Genes | Primer sequence (5'-3') | Product size (Bp) | Annealing temp | Reference |
|---|---|---|---|---|
| *bfpA* | F: AATGGTGCTTGCGCTTGCTGC | 324 | 60 | [24] |
| | R: CCGCTTTATCCAACCTGGTA | | | |
| *stx1* | F: AGTTAATGTGGTGGCGAA | 817 | 58 | [25] |
| | R:GACTTGCCGCTTCCAT | | | |
| *aggR* | F: GTATACACAAAGAAGGAAGC | 254 | 58 | [25] |
| | R: ACAGAATCGTCAGCATCAGC | | | |
| *LT* | F: GCACACGGAGCTCCTCAGTC | 218 | 60 | [25] |
| | R:TCCTTCATCCTTTCAATGGCTTT | | | |
| *IpaH* | F- AGGTCGCTGCATGGCTGGAA R- CACGGTCCTCACAGCTCTCA | 99 | 61 | [26] |

for aggregative adherence fimbria I), and EIEC (*ipaH* encoding invasion plasmid antigen H). All *E. coli* isolates and positive controls (*E. coli* ATCC 25922) for each of the genes tested had their DNA extracted using the boiling method as previously described [22]. PCR water was used as a negative control. The cycling conditions for all the PCR amplifications of pathotypes genes were as follows: 94°C for 5 minutes for initial denaturation and 30 cycles of 95°C for 30 seconds (denaturation), annealing at 62°C for 1 minute (for other primers) and 72°C for 1 minute 30 seconds (extension), followed by 72°C for 10 minutes (final extension); performed in 0.2 mL Eppendorf PCR tubes in a thermal cycler (MJ Research PTC-200). Amplified PCR products were separated and visualized by electrophoresis using 1% agarose tris-acetate-EDTA gel containing 0.5 μg EtBr staining. A 100–1,000 bp DNA ladder (Fermenters) was used to determine molecular weight.

**Antibiotic susceptibility testing.** The antimicrobial sensitivity profiles of all the isolated *E. coli* isolates were performed using the Kirby Bauer disk diffusion method against 12 antibiotics according to the Clinical and Laboratory Standards Institute guidelines [27]. There were seven drug classes tested: Beta-lactam, Amoxicillin + Clavulanic acid AMC (30 μg).; ampicillin AMP (10 μg); ceftriaxone CRO (30 μg); ceftazidime CAZ (30 μg), cefotaxime CTX(30 μg), cefpodoximine CPD(10 μg) Aminoglycosides (gentamicin CN(10 μg), Quinolones (nalidixic acid NA(30 μg); ciprofloxacin CIP(5 μg), Sulfonamides (trimethoprim-sulfamethoxazole STX (25 μg), Macrolides (azithromycin(15 μg), Phenicols (chloramphenicol CHL (30 μg). Tetracyclines (tetracycline TCY (30 μg) guidelines were provided by the Clinical and Laboratory Standards Institute (CLSI) in 2023. *E. coli* ATCC 25922 was used as a positive control. MDR was defined as resistance to at least one antibiotic from different antibiotic classes.

The zones of inhibitions were measured and interpreted using CLSI M100. Isolates that were resistant to 3$^{rd}$ cephalosporin were scored as ESBLs (Extended Spectrum Beta-lactamase).

**Screening for extended-spectrum β-lactamase.** Phenotypic methods were used for further confirmation of β-lactamase production for all the isolates showing resistance to 3$^{rd}$ generation cephalosporin, namely ceftazidime, cefotaxime, and ceftriaxone. A combination of clavulanic acid (10 μg) and ceftazidime (30 μg) was used. Both discs were placed on Muller Hinton agar plates which were earlier swabbed by respective cultures and incubated for 24 hours at 37°C. More than 5 mm increase in the zone diameter for ceftazidime-clavulanic acid was considered positive ESBL production.

ESBL detection and ESBL-associated genes ($bla_{CTX-M}$, $bla_{TEM}$, and $bla_{SHV}$) were detected using conventional PCR and previously described primers. **Table 2** [19]

**Data analysis.** Descriptive analysis was used to show the distribution of *E. coli* and *E. coli* pathotypes per samples. Inferential statistics, using Pearson's chi-square test, were applied to assess associations between: sample type and *E. coli* positivity, *E. coli* pathotype and detection rate, and sample type and antibiotic resistance. Antimicrobial-resistant patterns were interpreted and analyzed using the M 100 Clinical and Laboratory Standards Institute (CLSI 2022) and also analyzed using WHONET 2024 https://whonet.org/.

**Table 2. Resistance genes analyzed and the primers used.**

| Target Gene | Primer sequence (5'-3') | Product size bp | Annealing temp (°C) | References |
|---|---|---|---|---|
| $bla_{TEM}$ | F: GCGGA ACCCCTA TTTG | 1120 | 62 | [25] |
| | R: TCTAAAGTATATATGAGTAAACTTGGTCTGAC | | | |
| $bla_{CTX-M}$ | F: ATGTGCAGGACCAGTAARGTATGGC | 593 | 58 | [25] |
| | R:TGGGTRAARTARGTSACCAGAAYCAGCGG | | | |
| $bla_{SHV}$ | F:TTCGCCTGTGTATTATCTCTCCTG | 854 | 58 | [25] |
| | R:TTAGCGTTGCCAGTGTYTCG | | | |

**Ethical considerations.** Ethical approval was obtained from the institutional review board at Kenya Medical Research Institute; Scientific and Ethics Review Unit (KEMRI/SERU/CMR/P00240/4770). Verbal informed consent was sought from the street food vendors before buying the street foods.

## Results

### Distribution of *E. coli* and DEC in Street food and water samples

A total of 384 samples (wastewater, drinking water, and street food (fries, githeri and mandazi]) each 77, were collected and analyzed. Of these 384 samples, 62 (16.1%) were positive for *E. coli*. Majority (58.1%, 36) of the 62 *E. coli*-positive samples were wastewater samples, with drinking water constituting 25.8% of the *E. coli*-positive samples (**Table 3**). Additionally, DEC pathotypes were detected in 77.4% (48/62) of the *E. coli*-positive samples. Wastewater samples had the highest proportion (64.6%, 31/48) of DEC isolates, with no DEC being detected in French fries (**Table 3**).

### Distribution of DEC recovered from different samples collected

Five DEC pathotypes were detected, with ETEC being the most prevalent pathotype (69%, 33/48), followed by STEC 52% (25/48) **Fig 1**. ETEC was common across all samples while hybrid pathotypes were detected in 30 isolates. All five pathotypes were detected in wastewater isolates with ETEC being the most predominant pathotype. In drinking water and *mandazi,* four pathotypes were detected while *githeri* had three pathotypes, as shown in **Fig 2**.

### Hybrid pathogenic types detected

Of the 48 DEC isolates identified, 30 had a combination of multiple genes resulting in hybrid pathogenic types made of two, three and four pathotypes(S2). The most diverse hybrid pathotypes were found in *E. coli* isolates recovered from wastewater, as shown in **Fig 3**. The hybrid strains in githeri and mandazi samples were relatively low compared to other sample types. ETEC, STEC, and ETEC, EIEC were the most prevalent hybrid combinations with a prevalence of 23% (7/30) ETEC, STEC, EIEC had a prevalence of 20% (6/30). Among the hybrid DEC, ETEC was the most prevalent pathotype.

**Table 3. Distribution of *E. coli* and DEC from different samples.**

| Sample Type | *E. coli* -positive samples (*N*=62) n (%) | DEC-positive samples (*N*=48) n (%) |
|---|---|---|
| Wastewater | 36 (58.1) | 31 (64.6) |
| Drinking water | 16 (25.8) | 11 (22.9) |
| French fries | 1 (1.6) | 0 |
| Mandazi | 3 (4.8) | 2 (4.2) |
| Githeri | 6 (9.7) | 4 (8.3) |

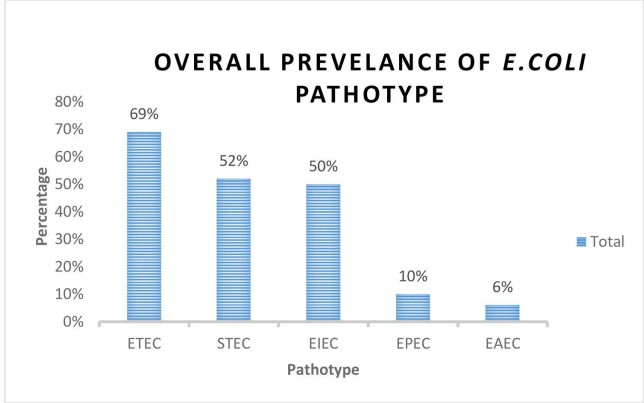

**Fig 1.** This graph illustrates the overall prevalence of *E. coli* pathotypes identified from the analyzed samples(Street food and water samples) in Mukuru Informal Settlement, Nairobi, Kenya.

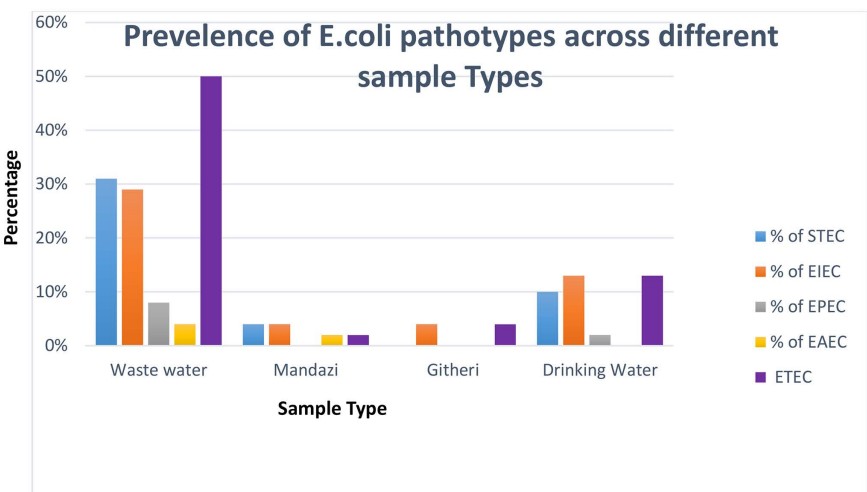

**Fig 2.** Presents the prevalence of *E. coli* pathotypes across different sample types collected in Mukuru informal settlements, Nairobi.

## Antimicrobial resistance profiles

The highest overall resistance of the 62 *E. coli* isolates was observed against tetracyclines (77.4%), sulfonamides (71.0%), and ampicillin (59.7%). Moderate resistance was observed to cefotaxime (50.0%), ciprofloxacin (41.9%), and azithromycin (37.1%). Low resistance rates were observed in kanamycin (16.1%), chloramphenicol (8.1%) and amoxicillin-clavulanic acid (4.8%) (**Table 4**).

## DEC resistance patterns to antibiotics

All 48 pathotypes exhibited varying levels of resistance to the antibiotics. ETEC exhibited high resistance rates to all the antibiotics compared to the other pathotypes, as shown in Fig 4. ETEC exhibited resistance to tetracycline (47.9%), trimethoprim-sulfamethoxazole (41.7%), cefotaxime (39.6%), and ampicillin (37.5%). STEC showed resistance to trimethoprim-sulfamethoxazole (45.8%), tetracycline (41.7%) ampicillin (35.4%). EAEC exhibited low

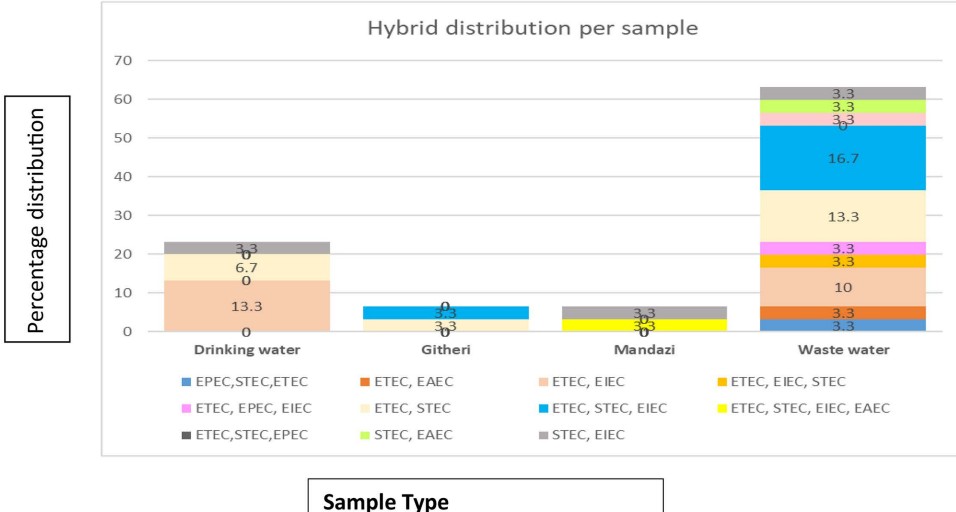

**Fig 3. Percentage distribution of hybrid *Escherichia coli* pathotypes across different sample types.**

**Table 4. Antibiotic resistance of *E.coli* isolated from the samples (*N*=62).**

| Drug | Drinking water | | Fries | | Githeri | | Mandazi | | Waste water | | Total | |
|------|------|------|------|------|------|------|------|------|------|------|------|------|
| | n | % | n | % | n | % | n | % | n | % | n | % |
| TCY | 13 | 21.0% | 1 | 1.6% | 2 | 3.2% | 3 | 4.8% | 29 | 46.8% | 48 | 77.4% |
| SXT | 11 | 17.7% | 1 | 1.6% | 3 | 4.8% | 3 | 4.8% | 26 | 41.9% | 44 | 71.0% |
| AMP | 11 | 17.7% | 1 | 1.6% | 2 | 3.2% | 2 | 3.2% | 21 | 33.9% | 37 | 59.0% |
| CTX | 8 | 12.9% | | | 2 | 3.2% | | | 21 | 33.9% | 31 | 50.0% |
| CIP | 10 | 16.1% | | | | | | | 16 | 25.8% | 26 | 41.9% |
| AZM | 6 | 9.7% | | | 3 | 4.8% | 1 | 1.6% | 13 | 21.0% | 23 | 37.1% |
| CRO | 7 | 11.3% | | | 2 | 3.2% | | | 13 | 21.0% | 22 | 35.5% |
| CPD | 6 | 9.7% | | | 1 | 1.6% | 1 | 1.6% | 12 | 19.4% | 20 | 32.3% |
| GEN | 6 | 9.7% | | | 1 | 1.6% | 1 | 1.6% | 12 | 19.4% | 20 | 32.3% |
| CAZ | 5 | 8.1% | | | 2 | 3.2% | | | 11 | 17.7% | 18 | 29.0% |
| NAL | 7 | 11.3% | | | | | | | 7 | 11.3% | 14 | 22.6% |
| KAN | 3 | 4.8% | | | | | 1 | 1.6% | 6 | 9.7% | 10 | 16.1% |
| CHL | 2 | 3.2% | | | | | | | 3 | 4.8% | 5 | 8.1% |
| AMC | | | | | | | | | 3 | 4.8% | 3 | 4.8% |

KEY: TCY -tetracycline, SXT -trimethoprim-sulfamethoxazole, AMP–ampicillin, CTX-cefotaxime, CIP –ciprofloxacin, AZM –azithromycin, CRO –ceftri-axone, CPD-cefpodoxime, GEN-gentamicin, CAZ –ceftazidime, NAL-nalidixic acid, KAN-kanamycin, CHL-chloramphenicol, AMC-amoxicillin-clavulanic acid.

resistance to ampicillin (6.3%), trimethoprim-sulfamethoxazole (6.3%), and tetracycline (6.3%); EPEC exhibited resistance to ampicillin (10.4%), trimethoprim-sulfamethoxazole (8.3%), and tetracycline (8.3%). EIEC had varying resistance rates to ampicillin (22.9%), trimethoprim-sulfamethoxazole (29.2%), and tetracycline (35.42%). EIEC did not exhibit resistance to amoxicillin-clavulanic acid, while EAEC did not exhibit resistance to azithromycin, chloramphenicol, and nalidixic acid.

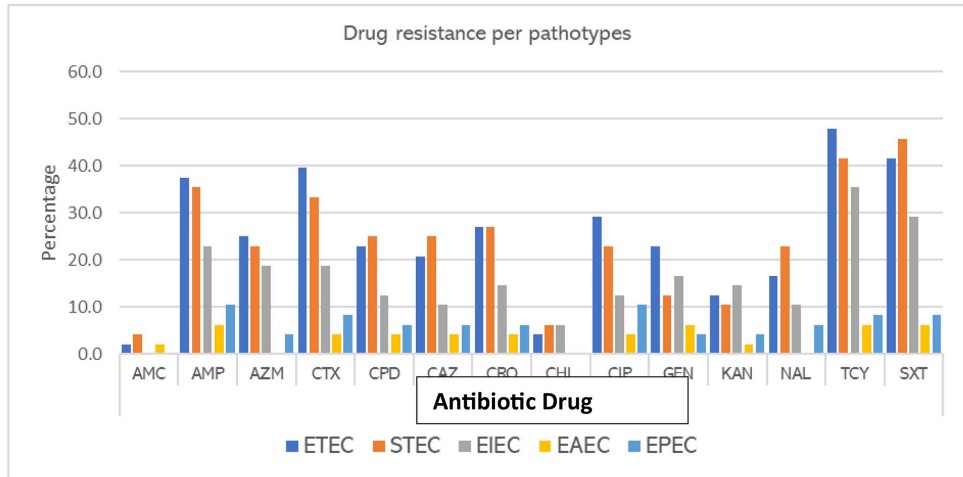

**Fig 4. Illustrates the percentage of drug resistance for various *E. coli* pathotypes against different antibiotic drugs from samples obtained in Mukuru informal Settlements,Nairobi,Kenya.**

**Association between sample type and antibiotic resistance.** Inferential analysis, using chi-square statistics, was applied to assess associations between sample type and antibiotics. *E. coli* isolates from wastewater had resistance to all the tested antibiotics (Table 5), with significant resistance to ampicillin (21/62 isolates), cefotaxime (21/62 isolates), and tetracycline (29/62 isolates). *E. coli* isolates from drinking water also had high resistance to tetracycline (13/62), ampicillin (11/62), trimethoprim-sulfamethoxazole (11/62), and ciprofloxacin (10/62). *E. coli* isolates from *Mandazi* and *githeri* had lower resistance overall but still exhibited resistance to beta-lactams, tetracycline, and sulfonamides. There was a significant difference in resistance to ciprofloxacin ($p=0.030$).

**Multidrug resistance in *E. coli*.** Majority 60% (27/45) of the MDR resistant *E.coli* isolates were recovered from wastewater, with 28.9% [13] from drinking water, 4.4% [2] from githeri, 4.4% [2] from mandazi and 2.2% [1] from French fries 2.2% [1] **Fig 5**. The frequency of resistance to 3, 4, 5, 6, and 7 classes of antimicrobial agents were 17.8% (8/45), 22.2% (10/45), 24.4% (10/45), 24.4% (10/45), 11.1% (5/45), respectively.

**Distribution of ESBL genes per sample.** Of the 62 *E. coli* isolates, 12.9% [8] carried $bla_{TEM}$ gene, of these, six were detected in wastewater while two in drinking water. The $bla_{CTXM}$ gene (S3) was detected in 3.2% [2] of the *E. coli* isolates; one in drinking water and one in wastewater The $bla_{TEM}$ gene was detected in multiple pathotypes, including ETEC, EAEC,

**Table 5. Association between sample type and Antibiotics.**

| Sample type | Antibiotic Resistance | | | | | | | | | | | | | |
|---|---|---|---|---|---|---|---|---|---|---|---|---|---|---|
| | AMC | AMP | AZM | CTX | CPD | CAZ | CRO | CHL | CIP | GEN | KAN | NAL | TCY | SXT |
| Drinking water | | 11 | 6 | 8 | 6 | 5 | 7 | 2 | 10 | 6 | 3 | 7 | 13 | 11 |
| French fries | | 1 | | | | | | | | | | | 1 | 1 |
| Githeri | | 2 | 3 | 2 | 1 | 2 | 2 | | | 1 | | | 2 | 3 |
| Mandazi | | 2 | 1 | | 1 | | | | | 1 | 1 | | 3 | 3 |
| Wastewater | 3 | 21 | 13 | 21 | 12 | 11 | 13 | 3 | 16 | 12 | 6 | 7 | 29 | 26 |
| **P-Values** | 0.533 | 0.849 | 0.937 | 0.359 | 0.699 | 0.608 | 0.619 | 0.956 | **0.030** | 0.731 | 0.582 | 0.059 | 0.172 | 0.561 |

TCY -tetracycline, SXT -trimethoprim-sulfamethoxazole, AMP–ampicillin, CTX-cefotaxime, CIP –ciprofloxacin, AZM –azithromycin, CRO –ceftriaxone, CPD-cefpodoxime, GEN-gentamicin, CAZ –ceftazidime, NAL-nalidixic acid, KAN-kanamycin, CHL-chloramphenicol, AMC-amoxicillin-clavulanic acid.

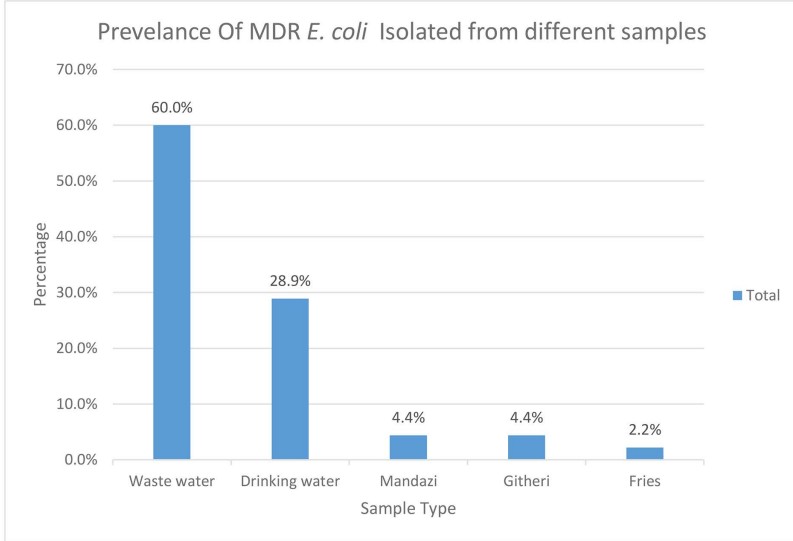

**Fig 5. Illustrates the percentage of Multi-Drug Resistant (MDR) *E. coli* detected across various sample types.**

STEC, EIEC, and EPEC. The $bla_{CTXM}$ gene was found in a nonpathogenic isolate and a hybrid isolate of ETEC, EPEC, and EIEC as shown in Table 6

## Discussion

Foodborne diseases caused by *E. coli* contamination are responsible for a huge public health challenge globally especially in low- and middle-income countries LMICs [1]. Understanding the role of environmental reservoirs in disseminating multidrug resistant diarrheagenic *E. coli* is crucial for mitigation strategies. This study was conducted in an urban informal settlement characterized by high population, dilapidated water, sanitation, and hygiene (WASH) infrastructure; therefore, most residents use common water points for drinking water and street foods are very popular within because they are relatively inexpensive, convenient, and easily accessible [28]. These conditions could contribute to a high burden of diarrheal morbidity within the population especially among children [19]. In this study, DEC were detected in 12.5% (48/384) of different samples with ETEC being the most prevalent pathotype. A huge proportion (72.6%, 45/62) of the *E. coli* isolates were observed to multi-drug resistant with high levels of resistance to ampicillin, tetracyclines and sulfonamides.

**Table 6. Distribution of ESBL genes Per Sample.**

| Sample Type | *E. coli* Pathotype | ESBL Gene |
|---|---|---|
| Wastewater | ETEC, EAEC | TEM |
| Wastewater | EPEC, STEC, ETEC | TEM |
| Wastewater | ETEC, STEC, EIEC | TEM |
| Wastewater | ETEC, STEC, EIEC | TEM |
| Wastewater | ETEC, STEC | TEM |
| Wastewater | ETEC, STEC, EPEC | TEM |
| Drinking water | ETEC, STEC | TEM |
| Drinking water | STEC | TEM |
| Drinking water | Negative | CTX-M |
| Wastewater | ETEC, EPEC, EIEC | CTX-M |

The detection of DEC in drinking water and food samples highlight the potential public health risks associated with these samples [29] in this setting. Wastewater had a high *E.coli* isolation rate compared to drinking water: similar findings were observed in a study conducted in Peru that showed drinking water samples had a lower prevalence of *E. coli* contamination, yet still posed a health concern [30] according to WHO standards, 2023. Although, 1% of *French fries* were *E. coli* positive this is concerning as these are ready to eat food and could easily result in foodborne infections. Detection of *E. coli* in street foods (*French fries*, *mandazi* and *githeri*) might be attributed to improper handling post-preparation or cross-contamination during storage, given that it is prepared in high temperatures capable of eliminating bacterial pathogens [31–33]. In this study, ETEC was the predominant DEC pathotype detected, these finding is comparable to findings observed in Kat River and Fort Beaufort abstraction water where ETEC was the most prevalent pathotype [34]. On the contrary, these findings were different from a study conducted in South Africa where diffusely adherent *E. coli* (DAEC) was the only pathotype isolated from wastewater [35]. Another study conducted in Johannesburg, South Africa, observed that EPEC was the dominant pathotype detected in environmental water samples [36].

The high prevalence of ETEC in environmental samples observed in this study is a huge public health concern, given that ETEC is a leading cause of bacterial diarrhea, particularly in children under five and travelers in low-resource settings [37]. The transmission of ETEC through contaminated food or water can cause outbreaks, especially where sanitation and hygienic conditions are poor. The presence of ETEC in water and food samples suggests widespread fecal contamination which highlights the poor water and sanitation conditions in the urban informal settlement. Hybrid DEC pathotypes were detected in this study which is comparable to previous findings observed in South Africa. Interestingly, one street food sample had 4 different DEC strains detected ETEC, STEC, EIEC and EAEC. The simultaneous presence of a combination of virulence factors in one isolate greatly increases the potential for infection and disease transmission among consumers [38]. A previous foodborne outbreak in Germany which spread out to Europe and North America was attributed to contamination of food samples by a hybrid DEC [39]. The presence of such hybrid DEC in Mukuru could pose a high risk of severe outbreaks, especially given the community's poor sanitation infrastructure.

Antimicrobial resistance in bacteria from food and water is a major threat to global public health [40] as it limits the management of foodborne diseases. In this study, high resistance to ampicillin, sulfamethoxazole and tetracycline was observed across all sample types. These results are similar to a study done in Korogocho, an urban slum in Nairobi, Kenya [41]. This observation suggests that environmental compartments in urban slums are plausible reservoirs of antibiotic resistant bacteria [42]. Resistance to ciprofloxacin showed a statistically significant association with sample type particularly in wastewater samples, which suggests that such environments might act as reservoirs for resistant bacteria [42]. The high resistance in *E. coli* recovered from wastewater observed in this study highlights the need for better waste management systems to prevent the spread of resistant bacteria.

*E. coli* isolates from *French fries* and *mandazi* displayed alarmingly complete resistance to tetracyclines and sulfonamides underscoring the increasing resistance. These findings align with studies in sub-Saharan Africa, where resistance to tetracyclines and sulfonamides is also consistently reported as high [43]. A study in Kenya by Omulo *et al.* found that over 80% of *E. coli* isolates from both human and environmental sources exhibited resistance to these antibiotic classes, similar to the trends observed in this study [41]. Additionally, studies from Nigeria and Tanzania reported comparable patterns, with tetracyclines and sulfonamides showing the highest resistance rates across environmental and food samples [44,45]. The findings on multidrug resistance among *E. coli* and its pathotypes is a significant public health concerns. Multidrug resistance was observed across nonpathogenic and pathogenic *E. coli* with isolates from wastewater demonstrating notably high resistance across several antibiotic classes.

Among the extended spectrum beta lactamase genes analysed, $bla_{TEM}$ was the most prevalent gene identified. These results are similar to other studies showing $bla_{TEM}$ and $bla_{CTXM}$ as the most prevalent ESBL genes with no isolation of $bla_{SHV.}$ [46]. This occurrence could likely be attributed to the widespread use of third- and fourth-generation cephalosporins. The high resistance to third-generation cephalosporins could lead to the clinical challenges in treating infections, especially

with the limited availability of alternative antibiotics in low-resource settings [46]. The high rates of multidrug resistance including the presence of extended-spectrum β-lactamase -producing *E. coli,* have serious implications that suggest widespread environmental contamination with antibiotic resistance genes, potentially due to the overuse or misuse of antibiotics in both human and animal populations.

In conclusion, street food and water samples in Mukuru serve as reservoirs for pathogenic *E. coli,* with the detection of diverse and hybrid pathotypes posing a public health concern, highlighting fecal contamination as a major route for the transmission of waterborne and foodborne pathogens. The presence of hybrid pathotypes, with varied virulence mechanisms and disease outcomes, complicates infections and treatment, particularly with the emergence of multidrug-resistant (MDR) and extended-spectrum beta-lactamase (ESBL)-producing strains resistant to key antibiotics like ampicillin, nalidixic acid, and trimethoprim-sulfamethoxazole. These findings highlight the need for enhanced surveillance and stricter regulations on antibiotic use in agricultural, clinical, and environmental sectors, particularly in low-resource settings. Addressing the One Health dimensions of AMR requires a coordinated approach to mitigate its spread across human, animal, and environmental interfaces.

## Supporting information

**S1 Fig. The Ingredients and description of street foods collected and analyzed from Mukuru informal settlement Nairobi Kenya.**
(ZIP)

## Acknowledgments

We would like to thank and acknowledge the Kenya Medical Research Institute (KEMRI) and Kenyatta University for infrastructure and laboratory reagents. We also acknowledge the field workers from the Center for Microbiology Research, KEMRI for collecting the samples.

## Author contributions

**Conceptualization:** Sheillah Mundalo.

**Data curation:** Sheillah Mundalo, Evans Kiptanui.

**Formal analysis:** Sheillah Mundalo.

**Funding acquisition:** Sheillah Mundalo.

**Investigation:** Sheillah Mundalo.

**Methodology:** Sheillah Mundalo, Susan Kavai.

**Project administration:** Sheillah Mundalo, Samuel Kariuki.

**Resources:** Sheillah Mundalo, Rael Too, Diana Imoli, Brian Silantoi, Cecilia Mbae.

**Software:** Sheillah Mundalo.

**Supervision:** Sheillah Mundalo, Regina Ntabo, Kelvin Kering, Samuel Kariuki, Cecilia Mbae.

**Validation:** Sheillah Mundalo.

**Visualization:** Sheillah Mundalo.

**Writing – original draft:** Sheillah Mundalo.

**Writing – review & editing:** Sheillah Mundalo, Regina Ntabo, Kelvin Kering, Rael Too, Kevin Kariuki, Diana Imoli, Brian Silantoi, Susan Kavai, Samuel Kariuki, Cecilia Mbae.

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
