## [Decision Letter · Decision Letter 0]

12 Feb 2025

Dear Dr. Mundalo,

Thank you for submitting your manuscript to PLOS ONE. After careful consideration, we feel that it has merit but does not fully meet PLOS ONE’s publication criteria as it currently stands. Therefore, we invite you to submit a revised version of the manuscript that addresses the points raised during the review process.

We look forward to receiving your revised manuscript.

Kind regards,

Babak Pakbin

Academic Editor

PLOS ONE

Journal requirements:   When submitting your revision, we need you to address these additional requirements. 1. Please ensure that your manuscript meets PLOS ONE's style requirements, including those for file naming. The PLOS ONE style templates can be found at https://journals.plos.org/plosone/s/file?id=wjVg/PLOSOne_formatting_sample_main_body.pdf and https://journals.plos.org/plosone/s/file?id=ba62/PLOSOne_formatting_sample_title_authors_affiliations.pdf. 2. Your abstract cannot contain citations. Please only include citations in the body text of the manuscript, and ensure that they remain in ascending numerical order on first mention. 3. PLOS requires an ORCID iD for the corresponding author in Editorial Manager on papers submitted after December 6th, 2016. Please ensure that you have an ORCID iD and that it is validated in Editorial Manager. To do this, go to ‘Update my Information’ (in the upper left-hand corner of the main menu), and click on the Fetch/Validate link next to the ORCID field. This will take you to the ORCID site and allow you to create a new iD or authenticate a pre-existing iD in Editorial Manager. 4. In your Methods section, please provide additional information regarding the permits you obtained for the work. Please ensure you have included the full name of the authority that approved the field site access and, if no permits were required, a brief statement explaining why. 5. Please amend either the abstract on the online submission form (via Edit Submission) or the abstract in the manuscript so that they are identical. 6. Please include captions for your Supporting Information files at the end of your manuscript, and update any in-text citations to match accordingly. Please see our Supporting Information guidelines for more information: http://journals.plos.org/plosone/s/supporting-information. 

Reviewers' comments:

Reviewer's Responses to Questions

**Comments to the Author**

1. Is the manuscript technically sound, and do the data support the conclusions?

Reviewer #1: Yes

Reviewer #2: Partly

Reviewer #3: Yes

2. Has the statistical analysis been performed appropriately and rigorously?

Reviewer #1: Yes

Reviewer #2: I Don't Know

Reviewer #3: Yes

3. Have the authors made all data underlying the findings in their manuscript fully available?

Reviewer #1: Yes

Reviewer #2: Yes

Reviewer #3: Yes

4. Is the manuscript presented in an intelligible fashion and written in standard English?

Reviewer #1: Yes

Reviewer #2: No

Reviewer #3: Yes

Reviewer #1: Dear Editor

While thanking you for your attention and inviting me to cooperate for intellectual participation in the field of reviewing and judging the manuscript entitled "Molecular Detection and Antibiotic Resistance of Diarrheagenic Escherichia coli from Street Food and Water in Mukuru slums, Nairobi County ", the suggested items to improve the scientific quality of the above article will be sent as follows:

1- Statistical analysis should be included in the abstract section of the article.

2- The formula for determining sample size should be stated.

3- The sources used in the methodology section must be included in the article.

4- Old resources should be replaced.

The following articles should be used in the introduction and discussion section:

*Prevalence, phylogroups and antimicrobial susceptibility of escherichia coli isolates from food products

*Antibiotic Resistance and Molecular Characterization of Cronobacter sakazakii Strains Isolated from Powdered Infant Formula Milk

*PREVALENCE OF SALMONELLA STRAINS ISOLATED FROM INDUSTRIAL QUAIL EGGS AND LOCAL DUCK EGGS, IRAN

* Antibiotic susceptibility and genetic relatedness of Shigella species isolated from food and human stool samples in Qazvin, Iran

*Genetic diversity and antibiotic resistance of Shigella spp. isolates from food products

* Genotypic and antimicrobial resistance characterizations of Cronobacter sakazakii isolated from powdered milk infant formula: A comparison between domestic and imported products

Reviewer #2: Manuscript ID: PONE-D-24-59070

Molecular Detection and Antibiotic Resistance of Diarrheagenic Escherichia coli

from Street food and Water in Mukuru slums, Nairobi County.

The manuscript titled "Molecular Detection and Antibiotic Resistance of Diarrheagenic Escherichia coli

from Street food and Water in Mukuru slums, Nairobi County" focuses on the research, both phenotypic and genomic characterization of E. coli strains in wastewater, drinking water and street food. The experimental approach adopted is not extremely innovative, but it allows the authors to collect numerous and important data.

However, I believe that the manuscript is not ready to be considered for publication because it has some important weaknesses described below.

Overall, the manuscript is not well written. The "Result" paragraph is extremely synthetic, so much so that it is not possible to understand why and how the results were calculated and expressed. The main weaknesses of the manuscript are Results and Discussion; furthermore, statistical analysis is sometimes mentioned in the text but there is no description of the analysis performed in "Materials and Method".

Abstract

Lines 21-23. The sum is not 100%, but 62.5. Please review the percentage calculations. Furthermore, and more seriously, there is no correspondence with the data reported in the text.

The discrepancy between the data reported in the various points of the text is frequent. In fact, for example, in the line 27 the authors say that tetracycline resistance is 70%, but in the text in the line 194 the authors say it is 100%. Since it is not clear when in the text the authors refer to isolated E. coli or to DEC, it is appropriate to underline this aspect.

Introduction

There are some acronyms that should be explained (line 43 and line 72).

Lines 38-46. The numeration of bibliography in the text should be checked (the order is not respected).

Materials and Methods

The materials and methods are concise, but not sufficiently clear; several information seemed to be missing. More in details, the missing points and necessary corrections are reported below.

Table 1. In this table are listed more that genes mentioned in the text.

Lines 129-137. Review the text. Punctuation is missing and sometimes sentences don't make sense. Insert the weight of antibiotics

Table 2: Insert in table R: and F:.

Results

Table 3: check the percentage of Samples positive for E. coli pathotypes per Mandazi that is 2.6% and not 2.7%. Also, what kind of statistical analysis was performed? In M&M section the authors only talked about descriptive analysis

Lines 179 – 180. How were these percentages calculated? It is not clear at all. Their sum is greater than 150% and there is no correspondence with figure 1. Furthermore, they do not correspond at all to the values reported in the abstract.

Lines 179 - 183. Furthermore, the percentage of hybrid strains should be mentioned in this section, and it should be specified whether hybrid exclusion is taken into account in the calculation of pathotype percentages. In short, it is not clear whether these percentages are calculated on 48 (total DEC strains) or on 16 (no hybrid DEC strains).

Lines 194 – 198. This section needs to be rewritten because the subject and complement are reversed in the sentence structure (antibiotics are resistant to antibiotics, not vice versa). Additionally, this section needs to include data on resistance to all E. coli, DEC and others, because there is confusion between the resistance rates reported in the abstract and the text.

Lines 205 – 207. Please define what kind of statistics were performed to evaluate the significance of the results obtained.

Lines 215 – 228. These paragraphs, like figures 4 and 5, should be completely revised since the concept of MDR is expressed in the resistance of the same bacterium to at least 3 pharmacological classes. From figure 5 for Fries and Mandazi it can be deduced that the strains are resistant to both SUL and TETRA, but the third pharmacological class involved is not intuitive and understandable. The text does not clarify anything about MDR strains, much less the figures that do nothing but represent the % of resistance to each of the pharmacological classes involved in the study.

Lines 232 – 233. In these lines, a further inconsistency is found between the data reported in the text and those in the table, in which 6 strains from wastewater and two from drinking water are positive for the tem gene.

Discussion

Lines 261 – 262. How were these percentages calculated? Furthermore, they are still different from those reported in the abstract

Overall, the discussion is very repetitive. The weight of DEC and AMR on human health is repeatedly emphasized and reiterated. The concept is correct, but it is simply repeated. Some percentages referring to the results are reinserted, but with different values that once again create confusion in the reader.

Figures

Fig. 1 The sum of the percentages is 99%. Furthermore, what does the 1.1% value reported on the orange bar of the wastewater samples (under 15.4) indicate?

Fig. 2 The sum of the percentages is 99.7%. However, if we are talking about hybrid why is ETEC (9.4%) reported in the wastewater bar?

Fig. 4 The figure can be avoided since it repeats data already listed in the results text. Also, together with figure 5, they do not represent MDR data, so they should be moved to other paragraphs.

Reviewer #3: Dear Author

The summary of the article and the introduction are well-written; the methodology and the results presented in the tables indicate the scope of the work and are well articulated, with a good discussion as well.

There are just two points that need clarification: firstly, considering the number of genes studied, why was a DNA extraction kit not used, and in the boiling method, the DNA purification is less efficient. Secondly, it would be beneficial to include an image of the multiplex gel, as it better illustrates the workflow.

Bacterial genera and species should be italicized in the text, tables, and references.

Regards

**Do you want your identity to be public for this peer review?** For information about this choice, including consent withdrawal, please see our Privacy Policy

Reviewer #1: No

Reviewer #2: No

Reviewer #3: No

---

## [Author Response · Author response to Decision Letter 1]

30 Apr 2025

Dear Reviewers,thank you for the thorough review you have done on my paper.It means alot to my career growth.i have attcahed all the responses in the response letter that has been attached.

---

## [Decision Letter · Decision Letter 1]

21 Jul 2025

Dear Dr. Mundalo,

Thank you for submitting your manuscript to PLOS ONE. After careful consideration, we feel that it has merit but does not fully meet PLOS ONE’s publication criteria as it currently stands. Therefore, we invite you to submit a revised version of the manuscript that addresses the points raised during the review process.

We look forward to receiving your revised manuscript.

Kind regards,

Babak Pakbin

Academic Editor

PLOS ONE

Journal Requirements:

Reviewers' comments:

Reviewer's Responses to Questions

**Comments to the Author**

Reviewer #4: (No Response)

Reviewer #5: All comments have been addressed

2. Is the manuscript technically sound, and do the data support the conclusions?

Reviewer #4: Partly

Reviewer #5: Yes

3. Has the statistical analysis been performed appropriately and rigorously?

Reviewer #4: No

Reviewer #5: Yes

4. Have the authors made all data underlying the findings in their manuscript fully available?

Reviewer #4: Yes

Reviewer #5: Yes

5. Is the manuscript presented in an intelligible fashion and written in standard English?

Reviewer #4: Yes

Reviewer #5: Yes

Reviewer #4: In this study, the authors investigated various characteristics of Escherichia coli isolated from food and street water, but the results were not statistically analyzed properly. Comments for improving the manuscript are included within the main text.

Reviewer #5: The manuscript presents a cross-sectional study assessing the occurrence, pathotyping, and antimicrobial resistance (AMR) profiles of diarrheagenic Escherichia coli (DEC) in street food and water samples collected from the Mukuru informal settlements in Nairobi, Kenya. The topic is timely and important, especially considering the growing public health threat posed by antimicrobial resistance and fecal contamination in low-resource urban settings.

Although I was not involved in the initial round of peer review for this manuscript, it appears that the authors have adequately addressed the concerns and suggestions raised during the initial review process. The revisions are thoughtful, and the manuscript has been significantly improved as a result. However, I’ve the following suggestions:

1. The figure captions and corresponding explanations are not adequate. Provide specific details corresponding to each figure.

2. Provide appropriate Axis titles for figures 2,3,4,6 and 7.

3. Streamline content in the Results and Discussion section to avoid redundancy and focus on key insights specific to this setting.

**Do you want your identity to be public for this peer review?** For information about this choice, including consent withdrawal, please see our Privacy Policy

Reviewer #4: No

Reviewer #5: No

---

## [Author Response · Author response to Decision Letter 2]

5 Aug 2025

Dear Reviewers,thank you for the througher review of my manuscript. I have been able to revise all the corrections given as attached in the letter and manuscript with track changes

---

## [Decision Letter · Decision Letter 2]

16 Dec 2025

Molecular Detection and Antibiotic Resistance of Diarrheagenic Escherichia coli

from Street food and Water in Mukuru slums, Nairobi County.

PONE-D-24-59070R2

Dear Dr. Mundalo,

We’re pleased to inform you that your manuscript has been judged scientifically suitable for publication and will be formally accepted for publication once it meets all outstanding technical requirements.

Kind regards,

Debdutta Bhattacharya

Academic Editor

PLOS One

Additional Editor Comments (optional):

Reviewers' comments:

Reviewer's Responses to Questions

**Comments to the Author**

Reviewer #4: All comments have been addressed

Reviewer #5: All comments have been addressed

2. Is the manuscript technically sound, and do the data support the conclusions?

Reviewer #4: Yes

Reviewer #5: Yes

3. Has the statistical analysis been performed appropriately and rigorously?

Reviewer #4: Yes

Reviewer #5: Yes

4. Have the authors made all data underlying the findings in their manuscript fully available?

Reviewer #4: Yes

Reviewer #5: Yes

5. Is the manuscript presented in an intelligible fashion and written in standard English?

Reviewer #4: Yes

Reviewer #5: Yes

Reviewer #4: (No Response)

Reviewer #5: The authors have submitted a significantly revised manuscript (PONE-D-24-59070R2) in response to the initial critiques. The authors have adequately addressed all the specific concerns I raised in the previous round of review, and the manuscript is now clear, and well-organized.

Summary of Previous Revisions Addressed:

1.Figure Captions and Explanations: The captions have been greatly expanded and now provide adequate detail and context, allowing the figures to be understood independently of the main text. This is a significant improvement. However, Fig 3 caption is still missing and can be added in the final draft.

2.Figure Axis Titles: Appropriate and descriptive axis titles have been added to Figures 2, 3, 4, 6, and 7, ensuring clarity in data presentation.

3.Streamlining Content: The authors have streamlined the Results and Discussion sections focusing the narrative on the unique findings specific to the Mukuru informal settlements setting. This has enhanced the manuscript’s focus and impact.

**Do you want your identity to be public for this peer review?** For information about this choice, including consent withdrawal, please see our Privacy Policy

Reviewer #4: No

Reviewer #5: No

---

## [Editor Report · Acceptance letter]

PONE-D-24-59070R2

PLOS One

Dear Dr. Mundalo,

I'm pleased to inform you that your manuscript has been deemed suitable for publication in PLOS One. Congratulations! Your manuscript is now being handed over to our production team.

Kind regards,

on behalf of

Dr. Debdutta Bhattacharya

Academic Editor

PLOS One